



# How sensitive are modeled contemporary subsea permafrost thaw and thickness of the methane clathrates stability zone in Eurasian Arctic to assumptions on Pleistocene glacial cycles?

Valentina V. Malakhova[1] and Alexey V. Eliseev[2,3]

[1]Institute of Computational Mathematics and Mathematical Geophysics, Siberian Branch of the Russian Academy of Sciences, Novosibirsk, Russia
[2]A.M. Obukhov Institute of Atmospheric Physics, Russian Academy of Sciences, Moscow, Russia.
[3]Kazan Federal University, Kazan, Russia.

*Correspondence to:* A.V. Eliseev (eliseev@ifaran.ru)

**Abstract.** Single–point simulations with a model for thermal state of subsea sediments driven by the forcing constructed from the ice core data show that the impact of initial conditions is lost after $\sim 100\,\mathrm{kyr}$. The time scales of temperature propagation in sediments and respective permafrost response are $\sim 10 - 20\,\mathrm{kyr}$ which is longer than the present interglacial. The timings of shelf exposure during oceanic regressions and flooding during transgressions are important for representation of sediment

thermal state and hydrates stability zone (HSZ). These timings should depend on the contemporary shelf depth (SD). During glacial cycles temperature at the top of sediments is a major driver of HSZ vertical boundaries change for SD of few tens of meters, while the pressure exerted by oceanic water becomes more important for larger SD. Thus, even the existence of HSZ and its disappearance might not be easily tied to oceanic transgressions and regressions.

## 1 Introduction

The most part of the contemporary methane clathrates (sometimes called as methane hydrates) at continental margins are believed to form during glaciations occurred in last several million years (Behseresht and Bryant, 2012). Their response to climate changes during warm epochs (interglacials) including the Holocene is not known with a sufficient certainty. In particular, there are controversial claims about origins of the recently measured high fluxes of methane from the marginal Arctic seas (Westbrook et al., 2009; Shakhova et al., 2010b, a; Semiletov et al., 2012; Berndt et al., 2014). While Shakhova et al. (2010b),

Shakhova et al. (2010a), and Semiletov et al. (2012) attribute the observed methane vents to the ongoing global warming, it might be related to the processes of much longer time scales, e.g., to the glacial cycles (Dmitrenko et al., 2011; Anisimov et al., 2014). This controversy has potentially important implications for the fate of methane clathrates during next several centuries (O'Connor et al., 2010).

    The latter controversy might be resolved, in principle, by means of numerical modeling of the thermal state of subsea

sediments and the state of methane clathrates during past glacial–interglacial cycles. This modeling, however, is uncertain itself. This uncertainty partly results from imprecise values of the model's governing parameters (e.g., thermophysical properties of the sediment column or intensity of the geothermal heat flux). This uncertainty was partly explored by Eliseev et al. (2015).





Another sources of uncertainty stem from imperfect specification of initial conditions and history of oceanic transgression and regression during glacial cycles.

With respect to the latter, some studies make unrealistic assumptions about oceanic regressions and transgressions during these cycles. Obviously, the timing of these regressions and transgressions should depend on the contemporary shelf depth (e.g., Bauch et al., 2001). According to direct measurements at the Eurasian Arctic shelf, subsea permafrost top is deeper for points with larger distance from the shoreline and, thus, which were flooded earlier during the last oceanic transgression (Overduin et al., 2015). However, for instance, Portnov et al. (2014) assume instantaneous exposition and flooding over their whole study area despite the corresponding range of contemporary shelf depths from zero to about $60\,\mathrm{m}$. Instantaneous flooding was assumed by Razumov et al. (2014) for the depth range from 4 to $100\,\mathrm{m}$. In addition, Denisov et al. (2011) and Anisimov et al. (2012) assume that permafrost, which was formed at shelf during the last glaciation, survives up to the present. In contrast, Nicolsky et al. (2012) account for the whole history of sea level changes during the last glacial cycle.

The impact of such assumptions is still unexplored. To study this, in the present paper single–point simulations are performed with a model of thermal state of subsea sediments. In order not to complicate the study by another source of uncertainty — availability of methane to form the clathrates (which is potentially very important as well) — we deal only with the hydrates stability zone (HSZ), i.e., the zone in which clathrates are thermodynamically stable whether they exist there or not.

## 2    Simulations with stratified shelf topology

We use the model for thermal state of subsea sediments, which was initially developed by Denisov et al. (2011) and further extended by Eliseev et al. (2015) and by Malakhova and Golubeva (2016). The model solves the one–dimensional equation for heat diffusion in sediment column subject to prescribed temperature at the sediment–ocean interface and prescribed heat flux at the bottom boundary. The depth of the latter boundary is set to $1,500\,\mathrm{m}$, and the prescribed value of the heat flux from the Earth interior is $G = 6 \times 10^{-2}\,\mathrm{W\,m^{-2}}$. More detailed model description is available at (Malakhova and Golubeva, 2016). Because this model is jointly developed by the A.M. Obukhov Institute of Atmospheric Physics, Russian Academy of Sciences (IAPRAS) and Institute of Numerical Mathematics and Mathematical Geophysics, Siberian Branch of the Russian Academy of Sciences (ICMMG), this model thereafter is referred to as IAPRAS–ICMMG sediment model (SM).

When shelf is covered by water, $T_\mathrm{B}$ is set to $-1.8^\circ\mathrm{C}$. During oceanic regression, this temperature is calculated as a sum of $T_\mathrm{r} + \Delta T_\mathrm{V}$. Here $T_\mathrm{r} = -12^\circ\mathrm{C}$, which is the present–day annual mean surface air temperature above the East Siberian Arctic shelf, and $\Delta T_\mathrm{V}$ is a time–varying anomaly derived from the Vostok ice core data as reported by Petit et al. (1999). Possible regional dependencies of temperature anomalies (Shakun et al., 2012) are ignored. The salinity of water in the sediment pores is considered to be the same as the oceanic one. As a result, the temperature of water freezing and melting is set to $T_\mathrm{f} = -1^o\mathrm{C}$. Initial temperature profile is set as a linear function from the vertical coordinate $z$, with the value at the upper boundary, which is equal to $-1.8^\circ\mathrm{C}$, and with the vertical gradient in the lower boundary, which continues the above mentioned value of the heat flux from the Earth interior. This linear profile is an equilibrium one with the specified boundary conditions provided that





heat diffusivity is independent from the vertical coordinate (Sharbatyan, 1974). However, because heat diffusivity depends on $z$, our simulations include a spin up of the length of $3\,\mathrm{kyr}$.

We performed single–point simulations for the last $400\,\mathrm{kyr}$ with different contemporary isobaths, $H_\mathrm{B}$, from 10 to $100\,\mathrm{m}$. As stated in the Introduction, these points differ between each other by timings when oceanic regressions and transgressions cross these depths. The timings of regressions and transgressions for each isobath for the last $400\,\mathrm{kyr}$ are prescribed according to (Waelbroeck et al., 2002).

During the first $\sim 100\,\mathrm{kyr}$ of the simulation, temperature adjusts to imposed boundary conditions. In contrast, at least two most recent major cycles do not show discernible inter–cycle trends, which is reflected in the values of temperature minima. The lack of these trends is supported by an additional simulation similar to just described, but with repeating the last $121.2\,\mathrm{kyr}$ of the forcing data several times (Fig. S1 in Supporting Information). In the latter simulation, the cycle–to–cycle variability is caused only by the intercycle variability of the imposed boundary conditions. In this model run, the third and later cycles are almost identical to each other, and the second cycle deviates from them only slightly.

In the original simulation, the coldest temperatures for the last two major glacial cycles differ between each other no more than by $0.3^\circ\mathrm{C}$ for the shallowest $H_\mathrm{B} = 10\,\mathrm{m}$ (no more than by $0.1^\circ\mathrm{C}$ for $z \geq 500\,\mathrm{m}$). For larger $H_\mathrm{B}$, this offset becomes higher, up to $2.5^\circ\mathrm{C}$ (but again no more than few degrees Centigrade for $z \geq 500\,\mathrm{m}$). For two most recent major cycles, the smaller $H_\mathrm{B}$ the colder temperature is in the depth of few hundred meters below the sediment top (Fig. 1). At these depths, impact of glacial cycles is most evident for intermediate $H_\mathrm{B}$. The latter is a result of mutually compensating interplay between the glacial and interglacial phases of these cycles.

Difference of timing for these minima between different vertical levels illustrates the characteristic time which is needed for heat wave to penetrate through the sediment column. Typically, minima occur by 15–20 thousand year later at depth $z = 1{,}500\,\mathrm{m}$ than at the top of the sediments. This time is very substantial. For instance, it is longer than the duration of most inter-glacials in the Pleistocene including the present interglacial, the Holocene (e.g., Past Interglacials Working Group of PAGES, 2016). As a result, during interglacial as well as during the substantial part of glaciation, temperature in the deep sediments still adjusts to the previous phase of the glacial–interglacial cycle.

As expected, permafrost base is shallower for larger $H_\mathrm{B}$. However, there are qualitative difference for evolution of this variable depending on $H_\mathrm{B}$. The depth of this base, $z_\mathrm{p}$ generally increases for $H_\mathrm{B} = 10\,\mathrm{m}$ during last glaciation from $489\,\mathrm{m}$ $\approx 155\,\mathrm{kyr\,B.P.}$ to $696\,\mathrm{m}$ $15.0 - 12.6\,\mathrm{kyr\,B.P.}$ (Fig. 1b). Such permafrost aggradation is simulated for all $H_\mathrm{B} \leq 30\,\mathrm{m}$. For intermediate $H_\mathrm{B}$, $50\,\mathrm{m}$ and $70\,\mathrm{m}$, $z_\mathrm{p}$ deepens during the first quarter and the first half of the last glaciation correspondingly despite of the warming at the top of sediments. Thereafter $z_\mathrm{p}$ begins to deepen until approximately the start of the Holocene (Fig. 1d). In the Holocene, permafrost base shallows, and the onset of this shallowing occurs earlier for larger $H_\mathrm{B}$. For the largest $H_\mathrm{B} = 100\,\mathrm{m}$, the permafrost aggregates until $27.2\,\mathrm{kyr\,B.P.}$ and degrades thereafter (Fig. 1f).

The qualitative difference with respect to $H_\mathrm{B}$ is simulated for the HSZ upper and lower boundaries, $z_\mathrm{HSZ,t}$ and $z_\mathrm{HSZ,b}$ respectively. For $H_\mathrm{B} \leq 30\,\mathrm{m}$, a thick continuous HSZ develops, which survives even during interglacials (Fig. 1b). For the last glacial cycle, its largest thickness is slightly above $1\,\mathrm{km}$ and is attained $12 - -13\,\mathrm{kyr\,B.P.}$ ($18 - -19\,\mathrm{kyr\,B.P.}$) for $H_\mathrm{B} = 10\,\mathrm{m}$ ($H_\mathrm{B} = 30\,\mathrm{m}$). In the Holocene, both $z_\mathrm{HSZ,t}$ and $z_\mathrm{HSZ,b}$ deepen. For the former, this deepening is expected due to oceanic





transgression. For the latter, it reflects an adjustment to the glacial intercycle variability. For larger $H_\mathrm{B}$, HSZ is absent during oceanic transgressions, and HSZ thickness is smaller (e.g., $\geq 680\,\mathrm{m}$ for $H_\mathrm{B} = 100\,\mathrm{m}$, see Fig. 1f). The glacial intercycle variability is more pronounced for larger shelf depths. For instance, the maximum HSZ thickness is $1090\,\mathrm{m}$ during the most recent major glacial cycle and $1085\,\mathrm{m}$ during the previous one for $H_\mathrm{B} = 10\,\mathrm{m}$ implying the cycle–to–cycle variability range

of 0.5%. For $H_\mathrm{B} = 50\,\mathrm{m}$ ($H_\mathrm{B} = 100\,\mathrm{m}$) the respective numbers are $1004\,\mathrm{m}$ and $1016\,\mathrm{m}$ ($679\,\mathrm{m}$ and $787\,\mathrm{m}$) implying the same range of 1% (16%).

Another important issue is a relation of forcing and response in dependence on $H_\mathrm{B}$. For shallow $H_\mathrm{B}$, $z_{\mathrm{HSZ,t}}$ closely follows $T_\mathrm{B}$. The depth of the HSZ bottom again follows $T_\mathrm{B}$ but with a delay of $10 - 20\,\mathrm{kyr}$. The smaller temperature at the top of sediment, the shallower is $z_{\mathrm{HSZ,t}}$ and, with an account for such delay, the deeper is $z_{\mathrm{HSZ,t}}$. For intermediate $H_\mathrm{B}$, short–term

disappearance of HSZ is simulated, e.g. around $300\,\mathrm{kyr}$ and $280\,\mathrm{kyr}$ ago (Fig. 1b). Duration of such periods is few kyr. It is interesting that these periods occur when sea level drops below the respective $H_\mathrm{B}$ with corresponding major drop in $T_\mathrm{B}$. As a result, for intermediate oceanic depths, changes in both $z_{\mathrm{HSZ,t}}$ and $z_{\mathrm{HSZ,t}}$ are caused by changes in pressure rather than changes in temperature, contrary to that it was for shallow $H_\mathrm{B}$. Pressure changes are the main driver for changes in the positions of HSZ vertical boundaries for large $H_\mathrm{B}$ as well.

Based on important differences between the simulations with different $H_\mathrm{B}$, we conclude that it is important to account for timings of the oceanic transgressions and regressions for realistic representation of temperature and depths of permafrost and HSZ boundaries. In addition, we state that, in general, adjustment time of the temperature signal in the sediments is as large as few tens of millennia, and at least one complete glacial cycle should be simulated for reach more or less realistic representation of the contemporary thermal state of the sediment column. More detailed analysis of this topic will pursued in the next Section.

In addition, we conclude that major driver of the response of the depths of HSZ vertical boundaries during glacial cycles depends on the contemporary oceanic depth. For $H_\mathrm{B}$ not larger than few tens meters, existence and movement of these boundary are caused by temperature changes at the top of sediments. For larger $H_\mathrm{B}$, in contrast, they are mainly caused by pressure changes.

## 3    Simulations with different initial conditions for the last glacial cycle

In this Section, sensitivity of the obtained results to prescription of initial conditions is explored. We use the same single–point model set up as it was in the previous Section, but run the model with different initial conditions. The performed experiments differ between each other by the starting time, initial temperature in the the sediments, and by details of sea level changes during simulations (Table 1). In all experiments, initial temperature profile in the sediment is prescribed according to the equilibrium linear function of vertical coordinate $z$ (see previous Section). We note that simulation S400 is identical to that used in the

previous Section. Simulations S120 and A120 are designed to study the importance of initial conditions at the beginning of the last glacial cycle for the simulation of the contemporary thermal state of the oceanic sediments and HSZ boundaries. While the inclusion of the simulations A025 and S120r0 might be considered as an odds with the conclusions from the previous Section, it is included to illustrate how the neglect of the details of the last major glacial cycle might affect the realism of the obtained




results. Thereafter, we consider simulation S400 as a reference one and compare other simulations with it. Because of the difference in initial conditions between the simulations listed in Table 1 and based on the results of the previous Section, we discard the first 1/3 of each simulation and discuss the results only for the last $80\,\mathrm{kyr}$ ($25\,\mathrm{kyr}$ for the A025 simulation).

The simulated of the permafrost lower boundary stratifies our model runs into two distinct groups. The first group consists of simulations S400, S120, and A120 with rather similar response of $z_\mathrm{p}$, $z_\mathrm{HSZ,t}$, and $z_\mathrm{HSZ,t}$ among them (Fig. 2). In this group, the most close correspondence is found between runs S400 and S120, with run A120 deviating from them for the first half of the simulation. The latter deviations become more pronounced at larger $H_\mathrm{B}$.

The second group is formed by simulations S120r0 and A020. These simulations are reasonably close to each other but deviate markedly from the simulations belonging to the first group. The most pronounced differences are again at intermediate and large $H_\mathrm{B}$. For instance, for $H_\mathrm{B} = 100\,\mathrm{m}$, $z_\mathrm{p}$ is few tens of meters for the simulations from the first group during $80 - 27\,\mathrm{kyr\,B.P.}$, and it is equal to several hundred meters for the simulations from the second group. In turn, hydrates stability zone disappears in the middle part of the last glacial cycles in the simulations from the first group but survives in the simulations from the second one.

At shallow $H_\mathrm{B}$, these differences are relatively small. Another prominent feature that $z_\mathrm{HSZ,t}$ (but not $z_\mathrm{HSZ,b}$ and $z_\mathrm{p}$) rather similar between all runs since the Last Glacial Maximum.

During glaciations, the equilibrium depth of the permafrost base may be estimated as $z_\mathrm{p,eq} = \kappa_\mathrm{f}\,(T_\mathrm{f} - T_\mathrm{B})\,/G \approx 7 \times 10^2\,\mathrm{m}$ (Sharbatyan, 1974), where thermal conductivity of frozen sediments $\kappa_\mathrm{f} = 2.2\,\mathrm{W\,m^{-1}\,K^{-1}}$. This value is close to maximum $z_\mathrm{p}$ numerically obtained for $H_\mathrm{B} = 10\,\mathrm{m}$ in simulations S400 and A120. For other simulations (all $H_\mathrm{B}$) the numerically obtained $z_\mathrm{p}$ is shallower than $z_\mathrm{p,eq}$ (for S120, the deviation is relatively small though) reflecting insufficient characterization of initial conditions in those simulations. In turn, for simulations S400 and A120 (and for some extent for S120), the numerically obtained $z_\mathrm{p}$ is again shallower than $z_\mathrm{p,eq}$ for intermediate and deep $H_\mathrm{B}$. This is linked to relatively short period of oceanic regression leading to substantially non–equilibrium dynamics of $z_\mathrm{p}$ at these $H_\mathrm{B}$. The latter again indicates a necessity to adequately represent the timings of oceanic regressions and transgressions during glacial cycles.

We conclude from this Section that it is important to simulate at least one glacial cycle to achieve a realistic representation of thermal and HSZ–related state of contemporary shelf sediments. In addition, the shortcomings of simulation S120r0 show that it is important to account for timings of oceanic regressions and transgressions during this cycle.

## 4 Conclusions

In this paper, we performed single–point simulations with the IAPRAS–ICMMG SM for thermal state of subsea sediments (Eliseev et al., 2015; Malakhova and Golubeva, 2016). The model was forced by idealised forcing constructed from changes in sea level and temperature of high latitudes constructed from the Vostok ice core data.

We found that one major glacial cycle is enough to lose the impact of initial conditions. This is consistent with typical time scale of propagation of temperature signal in the sediments, which is of the order $15 - 20\,\mathrm{kyr}$. Simulations of smaller length typically fall short for adequate representation of the present–day thermal state of subsea sediments and the position of HSZ



bottom. The contemporary HSZ top is realistically represented even in the simulation which is started around the Last Glacial Maximum though. This might be considered with an optimism because changes in HSZ top dominate over the changes in HSZ base in anthropogenically forced simulations for the 21st century (Eliseev et al., 2015). However, the obtained time scale of temperature signal propagation in the sediments is longer than the present interglacial, the Holocene. It supports the view

that the contemporary changes in sediments might be related to the adjustments to the last glacial cycle termination rather than to the ongoing change in the atmospheric climate, at least at large depth within the sediments which is in agreement with (Dmitrenko et al., 2011; Anisimov et al., 2014)). Such adjustment should lead to gradual upward movement of HSZ bottom with a corresponding release of methane. Provided that the latter methane is penetrated upward due to HSZ inhomogeneities and that HSZ top has a seasonal dependence, it might affect the release of methane from the upper part of HSZ (Berndt et al.,

2014). As a result, simulation of HSZ base is potentially important even for the anthropogenically forced climate change expected in the 21st century.

Another important ingredient of realistic representation of sediment thermal state and HSZ vertical boundaries is timings of shelf exposure during oceanic regressions and flooding during transgressions. These timings should be specified depending on the contemporary depth of oceanic shelf $H_B$. As it already stated in the Introduction, the latter is basically accounted in most

simulations but the former is ignored in some of them.

In addition, during glacial cycles temperature at the top of sediments is a major driver of change in position of HSZ vertical boundaries for $H_B$ of few tens of meters, while the pressure exerted by oceanic water becomes more important for larger $H_B$. This complicates the response of HSZ to changes in sea level accompanied by major changes in both $T_B$ and pressure. Therefore, even the existence of HSZ and its disappearance might not be easily tied to oceanic transgressions and regressions, and

HSZ might respond to changes in sea level in a counterintuitive way. An additional complication is due to the aforementioned thermal inertia of the sediment column. In particular, at the largest studied $H_B = 100\,\mathrm{m}$ subsea permafrost and HSZ disappear during major glaciations but persists through most interglacials. The latter is consistent with the known subsea permafrost in the Arctic shelf (e.g., Romanovskii and Hubberten, 2001; Overduin et al., 2015).

We note that the obtained results that glacial–interglacial changes of HSZ vertical boundaries at intermediate and large $H_B$

are primary caused by changes in pressure imposed by oceanic water do not imply that the same is true for the anthropogenically–forced climate changes expected in the 21st century and, probably, beyond. The reason for this is due to large, typically by two orders of magnitude difference in sea level change during the Pleistocene glacial cycle which are of the order of one hundred meters (Masson-Delmotte et al., 2013; Past Interglacials Working Group of PAGES, 2016) on one side, and those expected in the incoming few centuries with characteristic values of tens of centimeters or few meters under state–of–the–art forcing sce-

narios (Church et al., 2013). In particular, as it was found by Buffett and Archer (2004) and Hunter et al. (2013), temperature changes dominate over pressure ones in HSZ response to anthropogenically–induced climate change during the 21st century.

Our paper has a number of caveats. The first one is that we ignore possible regional dependencies of temperature anomalies (Masson-Delmotte et al., 2013, and references therein) including the timings of rapid temperature changes (Shakun et al., 2012). In addition, the forcing data used in this paper are superceded by a newer data, e.g., from the EPICA ice drilling program



(EPICA Community Members, 2004). However, we consider both shortcoming as relatively unimportant because of idealized nature of our study and the obtained results are correct at least in a qualitative sense.

*Author contributions.* All calculations are made by V.V. Malakhova. Both authors equally contributed to the formulation of the goals of research and to the preparation of the text.

5 *Acknowledgements.* The authors are grateful to I.I. Mokhov for his insightful comments on the obtained results during preparation of the manuscript. This work has been supported by the Russian Foundation for Basic Research (grants 15–05–02457 and 15–35–21061). The part related to the results on subsea permafrost was supported by the Russian Science Foundation (grant 14–17–00647).



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



**Table 1.** Model runs to study the dependence on of the obtained results on initial conditions

| run | starting time, kyr B.P. | $T_{\mathrm{B}}(0)$, °C | sea level |
|---|---|---|---|
| S400 | 400 | -1.8 | from the Vostok ice core data |
| S120 | 120 | -1.8 | identical to S400 for last 120 kyr |
| A120 | 120 | -13.0 | identical to S400 for last 120 kyr |
| S120r0 | 120 | -1.8 | similar to S400 for last 120 kyr, but the oceanic regression occurs at 117 kyr B.P. irrespective of $H_{\mathrm{B}}$ |
| A025 | 25 | -21.0 | identical to S400 for last 25 kyr |

$T_{\mathrm{B}}(0)$ stands for initial temperature at the sediments top. The same value is applied for all $H_{\mathrm{B}}$.

Oceanic regression timing in S120r0 corresponds to the value for $H_{\mathrm{B}} = 10$ m



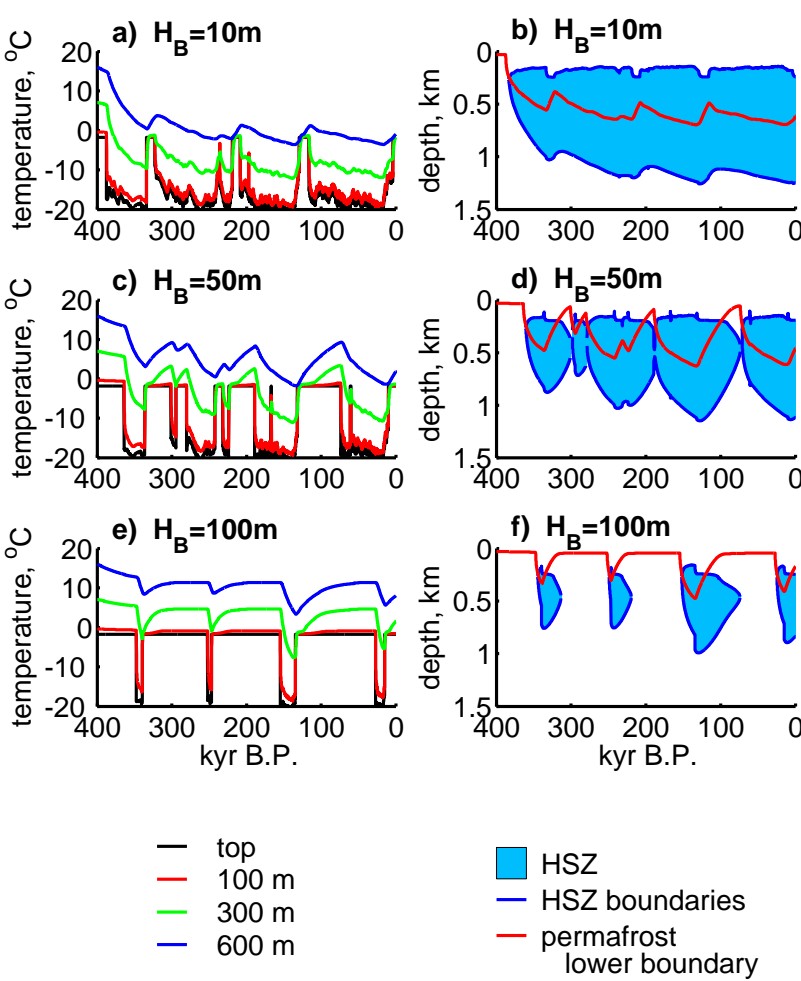

**Figure 1.** Temperature at different vertical levels in sediments (a, c and e) and depths (below the sediment top) of HSZ boundaries and permafrost bottom (b, d and f) in simulations with the contemporary sediment depths 10 m (a, b), 50 m (c, d) and 100 m (e, f).



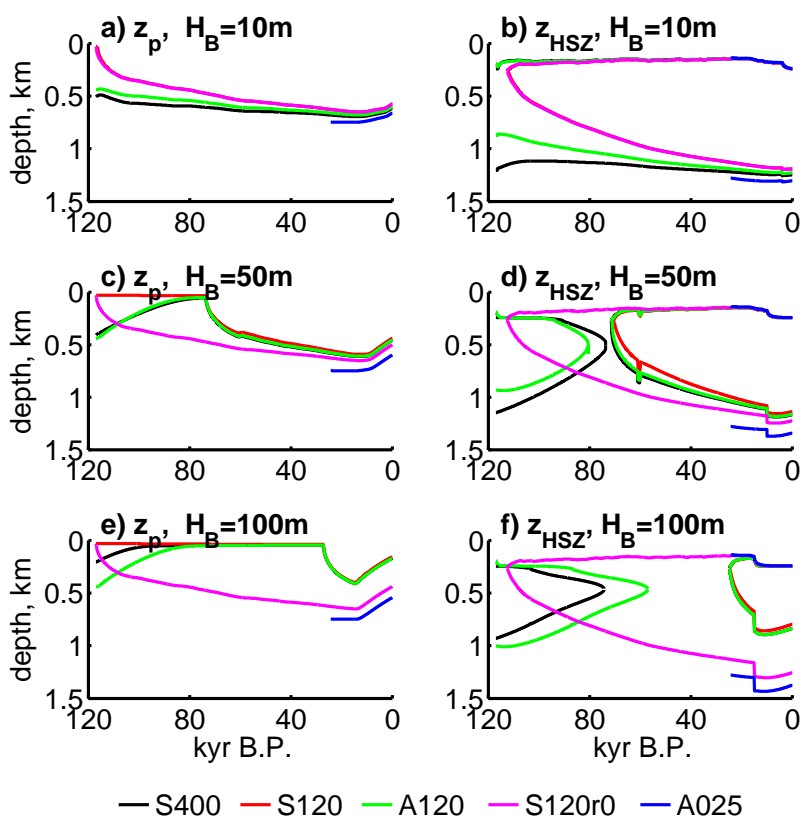

**Figure 2.** Depths (below the sediment top) of the permafrost bottom (a, c and e) and HSZ boundaries and (b, d and f) in simulations with the contemporary sediment depths 10 m (a, b), 50 m (c, d) and 100 m (e, f).