# Peer review of "How sensitive are modeled contemporary subsea permafrost thaw and thickness of the methane clathrates stability zone in Eurasian Arctic to assumptions on Pleistocene glacial cycles?"

_Climate of the Past, 2016_

## Referee Comment (RC1) · Anonymous Referee #1 · 5 Oct 2016

The paper presents a sensitivity study of the HSZ to details of sea level change and the consequent incursion and retreat of shelf sea inundation over the continental East Siberian Continental shelf. The shallow seas are exposed to the cold atmosphere for longer and hence form a deep permafrost layer. Depths at which the permafrost melts during each interglacial result in temporary hydrate stability zones, and I would presume low reservoirs of hydrates. The message is that time lags in ablation of the hydrate stability zone suggest that release of hydrate is a long process, and the timescale

of anthropogenic warming is comparatively short. The paper is poorly written, the logic is hard to follow. The paper fails to present the science-driven issue – what is currently understood and why is this particular study advancing the field. What is expected of the study and how does it achieve its ends. Clearly this is associated with the initial conditions of anthropogenic climate change and the rate of future emissions such as described in the papers below. All these papers make use of a sediment model for the HSZ to the model itself is not unique. The authors do recognise this and so do not use much text to describe it. Even through it may not be the best model available, it is adequate for the task to which it is applied – namely a sensitivity study. Kretschmer, K., A. Biastoch, L. Rüpke, and E. Burwicz (2015), Modeling the fate of methane hydrates under global warming. Global Biogeochem. Cycles, 29, 610–625. doi: 10.1002/2014GB005011. Marín-Moreno, H., T. A. Minshull, G. K. Westbrook, B. Sinha, and S. Sarkar (2013), The response of methane hydrate beneath the seabed offshore Svalbard to ocean warming during the next three centuries, Geophys. Res. Lett., 40, 5159–5163, doi:10.1002/grl.50985. The paper is rather to qualitative in that it does not attempt to quantify the uncertainties associated with the timing of the ocean inundations. It could perhaps use a metric of the volume of the HSZ as a function of the phase of the inundation. The writing of the paper still retains Russian phraseology with missing English definite article ('the') and indefinite articles ('a' and 'an'). I have suggested some changes for the introduction below, but the issue of confused logic and linguistics is common throughout, and would require major effort to provide corrections.

Detailed comments This title need to be shortened. Perhaps - "The stability of contemporary Eurasian Arctic methane hydrates to Pleistocene glacial cycles". NOTE: you use 'hydrates' in the abstract and HSZ throughout so the title should use the same nomenclature Page 1. Line 1-8. The abstract needs to start with a sentence of context. What is the scientific question being addressed and why is this important? i.e. "Why should I read this paper?" Line 1-2. Single–point simulations with a model , describing the thermal state of subsea sediments driven by the forcing constructed from ice core

data, show that the impact of the initial sediment conditions is lost after âĹij 100 kyr. Line 3-5. The timings of continental shelf exposure during oceanic regressions, and flooding during transgressions, are important for the representation of the sediment thermal state and hydrate stability zone (HSZ). Line 11. Replace 'in last' with 'over the last' Line 12. Replace "is not known with a sufficient certainty" with "is not well known" or otherwise you need to quantify "sufficient certainty" Line 13. Rather than addressing a "controversy" it might be better if the paper were presented as "reducing uncertainty", rather than about taking sides in a controversy. I thus suggest replacing "there are controversial claims about origins" with "there is uncertainty about the origins" Line 17. I suggest "The uncertainty in the driver of methane release has important implication for the release rate from hydrates over the coming centuries (O'Connor et al., 2010)" Line 19. I suggest "Understanding may be improved, in principle, by means of . . . . . ." page-1 Line 20 to page 2 Line2. I suggest "Such modelling contains its own uncertainties associated with its parameterisations (e.g. ….) as explored by Eliseev et al (2015), and initial state originating from sea level changes during glacial cycles."

Page 2. Line 3. Replace "assumptions" with "approximations" Line 4. 'Obviously' is redundant, start with "The timing of the regressions and transgressions depends on the contemporary shelf depth" Lines 7-11. Poorly posed. I suggest "Previous studies either assume instantaneous exposition and flooding over the entire shelf (Portnov et al., 2014; Razumov et al., 2014), or that the permafrost, formed during the last glaciations, persists up to the present (Denisov et al., 2011; Anisimov et al., 2012). Line 12. Suggest "The impact of such approximations on the presence of the hydrate stability zone (HSZ), the region of the subsea sediments in which hydrates are thermodynamically stable, regardless of their presence or not, is still unexplored. With this purpose we undertake a series of one-dimensional simulations using a model describing the thermal state of subsea sediments."

---

## Referee Comment (RC2) · Anonymous Referee #2 · 18 Jan 2017

Problem of HSZ stability at Arctic shelf during glacial-interglacial cycles was studied in this paper with a single column model. Paper is interesting, original, appropriate for the journal and can be published after major revision.

General comments: I am agree with Referee 1 that main deficiency of the paper is that the aim of the study, as well as main goal aren't formulated accurately. Also in the introduction one can't find exact formulation of what is known and what is not known about the subject under consideration.

Specific comments: One of main conclusions in the paper is that for HB not larger than tens meters temperature change is main driver for the changes of HSZ boundaries, while pressure change is crucial for deeper HB. This conclusion seems improbable. For example, at 600 meters increase of pressure by 10 atm (100 meters of water column) should produce the same effect as decrease of temperature by approximately 2 K according to curve of methane hydrate stability. But figure 1e shows that temperature change at 600m is as large as 5 degrees and should produce larger effect. At 300 meters, increase of pressure by 10 atm produce the same effect as cooling by 4 degrees, but fig.1d show cooling by 5-10 degrees. The seeming coincidence of maximum HSZ extension and maximum sea level during interglacials shown in fig.1f can be explained by delay of cooling wave with increase in depth. So, categorical statement that increase of pressure rather than cooling is a primary source of increase of HSZ volume for deep HB should be removed from abstract, conclusion and the end of chapter 2. It would be useful if authors present figure similar to their Fig.1f (and may be 1d, i.e. for HB=50m) but for experiment with prescribed change of pressure only with surface temperature fixed at -1.8C during 400 kyrs.

Minor comments: P.2, line 25-32. Why TB =-1.8C is not the same as Tf=-1C? This point should be explained.

P.2, line 32. Coefficient for specified initial linear temperature profile in K/m should be presented in the paper.

---

## Author Comment (AC1) · 1 Feb 2017

We are grateful for the reviewer for constructive comments which led to the improved presentation of our results.

The most important changes in the manuscript are as follows:

- The paper title is changed to 'The stability of contemporary subsea permafrost and associated methane hydrates to Pleistocene glacial cycles'. This is done in

order to shorten the title and emphasise that we study the hydrates which are formed during the Pleistocene cold epochs.

- We revised the abstract to the paper as well as sections *Introduction* and *Conclusions* to highlight our motivation to undertake this study. In particular, we show that, while it is widely acknowledged that the response of the shelf sediments to imposed oceanic warming is a slow process, the time scale of such response is not yet quantified. Some references are added with respect to this discussion. We show that this time scale is of the order of 10-20 kyr for the deep present–day shelf, which is as twice as large in comparison to similar estimates obtained by Romanovskii et al. (2005).

- To highlight our results for the time scales of the response of the sediment thermal state to temperature changes at the ocean–sediment interface, we extended our paper by new Fig. 2, which shows the lag of the HSZ thickness $D_{\text{HSZ}}$ with respect toÂă$T_{\text{B}}$, and by the paragraph devoted to the discussion of this Figure. Previous Fig. 2 is now referred to as Fig. 3.

- In response to the comment made by the second reviewer, we extended our paper by supplementary Fig. S2, which shows the results of additional simulations in which impacts of the pressure changes due sea level change are neglected.

- The language is checked and ameliorated.

- In addition, we discovered and corrected a technical error for our output for $z_{\text{HSZ,t}}$ (Figs. 1, 3, and S1). This error does not affect the major outcome of our manuscript.

Below, the point-to-point replies to the comments are listed. Original reviewer's comments are typed in italic.

**General comments**

*The paper is poorly written, the logic is hard to follow. The paper fails to present the science-driven issue – what is currently understood and why is this particular study advancing the field. What is expected of the study and how does it achieve its ends. Clearly this is associated with the initial conditions of anthropogenic climate change and the rate of future emissions such as described in the papers below. All these papers make use of a sediment model for the HSZ to the model itself is not unique. The authors do recognise this and so do not use much text to describe it. Even through it may not be the best model available, it is adequate for the task to which it is applied – namely a sensitivity study. Kretschmer, K., A. Biastoch, L. Rüpke, and E. Burwicz (2015), Modeling the fate of methane hydrates under global warming. Global Biogeochem. Cycles, 29, 610–625. doi: 10.1002/2014GB005011. Marín-Moreno, H., T. A. Minshull, G. K. Westbrook, B. Sinha, and S. Sarkar (2013), The response of methane hydrate beneath the seabed offshore Svalbard to ocean warming during the next three centuries, Geophys. Res. Lett., 40, 5159–5163, doi:10.1002/grl.50985. The paper is rather to qualitative in that it does not attempt to quantify the uncertainties associated with the timing of the ocean inundations. It could perhaps use a metric of the volume of the HSZ as a function of the phase of the inundation. The writing of the paper still retains Russian phraseology with missing English definite article ('the') and indefinite articles ('a' and 'an'). I have suggested some changes for the introduction below, but the issue of confused logic and linguistics is common throughout, and would require major effort to provide corrections.*

We are grateful to the reviewer for this very stimulating comment. It guided us how to express our motivation more clearly and how to emphasise the novelty of our study. In response to this general comment, we revised our manuscript as follows:

- In the abstract, as well as in *Introduction* we highlight our motivation to undertake this study. In particular, we show that, while it is widely acknowledged that the

response of the shelf sediments to imposed oceanic warming is a slow process, the time scale of such response is not yet quantified. In most previous papers the length of the performed simulations is up to few millennia which is not sufficient for such quantification (some references are added with respect to this discussion). The only exception, which we aware of, is the paper (Romanovskii et al., 2005) who also performed the simulations covering the whole glacial cycle. They obtained the time scale of the response of the subsea permafrost and of the subsea hydrates developed in this permafrost, which is of the order of 5–10 kyr.

- In sect. 2, we show that this time scale is of the order of 10–20 kyr for the deep present-day shelf, which is as twice as large in comparison to similar estimates obtained by Romanovskii et al. (2005). The likely reason of the latter difference are site–specific, non–monotonic profiles of the sediments thermophysical properties employed in the Romanovskii et al.'s paper. This, in principle, may diminish the generality of the conclusions of that paper.

- In *Introduction* and in *Conclusions* we stress that we do not attempt to estimate release of methane from hydrates which dissociate due to heat propagation into the sediment column from above during interglacials. A biogeochemical model explicitly simulating methane geochemistry in the sediments and formation and dissociation of methane hydrates should be used to pursue this goal. Such models need detailed (vertically resolved and, thus, tied to particular geographic locations) input sets though. The latter precludes to perform idealised, easier to interpret study as that presented here. Moreover, this modelling has its own uncertainties, e.g. due to the model structure, which are beyond the scope of the present manuscript. In particular, some of these models ignore hydrates in the subsea permafrost formed during Pleistocene glaciations. A simplified approach for assessment of hydrate methane volume could be an implementation of the transfer functions for hydrate stock as suggested by the reviewer. However, this approach is not pursued in our paper, because our goal is to study the hydrates

developed during the Pleistocene glacial cycles, while such transfer functions ignore these hydrates.

- In the abstract, in *Introduction* and in *Conclusions*, we highlight that our study is focussed on the hydrates in the subsea permafrost. These hydrates are distinct from those occurring, e.g., near Svalbard, where permafrost is not necessary for formation of hydrates. As a result, our results can not be directly compared with those reported by Marín–Moreno et al. (2013, 2015).

**Detailed comments**

- *This title need to be shortened. Perhaps - "The stability of contemporary Eurasian Arctic methane hydrates to Pleistocene glacial cycles".*
  The title is changed to 'The stability of contemporary subsea permafrost and associated methane hydrates to Pleistocene glacial cycles'.

- *NOTE: you use 'hydrates' in the abstract and HSZ throughout so the title should use the same nomenclature*
  Upon revision, all instances of word 'clathrates' are replaced by 'hydrates'. The only instance, where word 'clathrates' is kept in the paper is in the first sentence of the Introduction, in which, however, it is used only to explain the term 'hydrates'.

- *Page 1. Line 1-8. The abstract needs to start with a sentence of context. What is the scientific question being addressed and why is this important? i.e. "Why should I read this paper?*
  The abstract is extended by sentences discussing our motivation to undertake the study.

- *Line 1-2. Single–point simulations with a model , describing the thermal state of subsea sediments driven by the forcing constructed from ice core data, show that*

*the impact of the initial sediment conditions is lost after 100 kyr.*
The sentence is revised.

- *P. 1, LL. 3-5. The timings of continental shelf exposure during oceanic regressions, and flooding during transgressions, are important for the representation of the sediment thermal state and hydrate stability zone (HSZ).*
The phrase is ameliorated.

- *P. 1, L. 11. Replace 'in last' with 'over the last'*
The sentence is changed accordingly.

- *P. 1, L. 12. Replace "is not known with a sufficient certainty" with "is not well known" or otherwise you need to quantify "sufficient certainty"*
The phrase is ameliorated.

- *P. 1, L. 13. Rather than addressing a "controversy" it might be better if the paper were presented as "reducing uncertainty", rather than about taking sides in a controversy. I thus suggest replacing "there are controversial claims about origins" with "there is uncertainty about the origins"*
Words 'controversy' and 'controversial' is replaced by 'uncertainty' in this and other instances in the text.

- *P. 1, L. 17. I suggest "The uncertainty in the driver of methane release has important implication for the release rate from hydrates over the coming centuries (O'Connor et al., 2010)"*
The phrase is revised according he reviewer's suggestion.

- *P. 1, L. 19. I suggest "Understanding may be improved, in principle, by means of . . ."*
The sentence is revised.

- *P. 1, L. 20 to P. 2, L. 2. I suggest "Such modelling contains its own uncertainties associated with its parameterisations (e.g. . . ..) as explored by Eliseev et al (2015), and initial state originating from sea level changes during glacial cycles."*
According to the reviewer's suggestion, these phrases are merged together.

- *P. 2, L. 3. Replace "assumptions" with "approximations"*
The word 'assumption' us replaced by word 'approximation'.

- *P. 2, L. 4. 'Obviously' is redundant, start with "The timing of the regressions and transgressions depends on the contemporary shelf depth"*
The sentence is shortened.

- *P. 2, LL. 7-11. Poorly posed. I suggest "Previous studies either assume instantaneous exposition and flooding over the entire shelf (Portnov et al., 2014; Razumov et al., 2014), or that the permafrost, formed during the last glaciations, persists up to the present (Denisov et al., 2011; Anisimov et al., 2012).*
These phrases are ameliorated.

- *P. 2, L. 12. Suggest "The impact of such approximations on the presence of the hydrate stability zone (HSZ), the region of the subsea sediments in which hydrates are thermodynamically stable, regardless of their presence or not, is still unexplored. With this purpose we undertake a series of one- dimensional simulations using a model describing the thermal state of subsea sediments."*
The statements are revised according to the reviewer's suggestion.

---

## Author Comment (AC2) · 1 Feb 2017

article

We are grateful for the reviewer for constructive comments which led to the improved presentation of our results.

The most important changes in the manuscript are as follows:

- The paper title is changed to 'The stability of contemporary subsea permafrost and associated methane hydrates to Pleistocene glacial cycles'. This is done in order to shorten the title and emphasise that we study the hydrates which are formed during the Pleistocene cold epochs.

- We revised the abstract to the paper as well as sections *Introduction* and *Conclusions* to highlight our motivation to undertake this study. In particular, we show that, while it is widely acknowledged that the response of the shelf sediments to imposed oceanic warming is a slow process, the time scale of such response is not yet quantified. Some references are added with respect to this discussion. We show that this time scale is of the order of 10-20 kyr for the deep present–day shelf, which is as twice as large in comparison to similar estimates obtained by Romanovskii et al. (2005).

- To highlight our results for the time scales of the response of the sediment thermal state to temperature changes at the ocean–sediment interface, we extended our paper by new Fig. 2, which shows the lag of the HSZ thickness $D_{\mathsf{HSZ}}$ with respect to $T_B$, and by the paragraph devoted to the discussion of this Figure. Previous Fig. 2 is now referred to as Fig. 3.

- We extended our paper by supplementary Fig. S2, which shows the results of additional simulations in which impacts of the pressure changes due sea level change are neglected.

- The language is checked and ameliorated.

- In addition, we discovered and corrected a technical error for our output for $z_{\mathsf{HSZ,t}}$ (Figs. 1, 3, and S1). This error does not affect the major outcome of our manuscript.

Below, the point-to-point replies to the comments are listed. Original reviewer's comments are typed in italic.

**General comments**

*I am agree with Referee 1 that main deficiency of the paper is that the aim of the study, as well as main goal aren't formulated accurately. Also in the introduction one can't find exact formulation of what is known and what is not known about the subject under consideration.*

- In the abstract, as well as in *Introduction* we highlight our motivation to undertake this study. In particular, we show that, while it is widely acknowledged that the response of the shelf sediments to imposed oceanic warming is a slow process, the time scale of such response is not yet quantified. In most previous papers the length of the performed simulations is up to few millennia which is not sufficient for such quantification (some references are added with respect to this discussion). The only exception, which we aware of, is the paper (Romanovskii et al., 2005) who also performed the simulations covering the whole glacial cycle. They obtained the time scale of the response of the subsea permafrost and of the subsea hydrates developed in this permafrost, which is of the order of 5–10 kyr.

- In sect. 2, we show that this time scale is of the order of 10–20 kyr for the deep present-day shelf, which is as twice as large in comparison to similar estimates obtained by Romanovskii et al. (2005). The likely reason of the latter difference are site–specific, non–monotonic profiles of the sediments thermophysical properties employed in the Romanovskii et al.'s paper. This, in principle, may diminish the generality of the conclusions of that paper.

**Specific comments**

*One of main conclusions in the paper is that for HB not larger than tens meters temper-*

*ature change is main driver for the changes of HSZ boundaries, while pressure change is crucial for deeper HB. This conclusion seems improbable. For example, at 600 meters increase of pressure by 10 atm (100 meters of water column) should produce the same effect as decrease of temperature by approximately 2 K according to curve of methane hydrate stability. But figure 1e shows that temperature change at 600m is as large as 5 degrees and should produce larger effect. At 300 meters, increase of pressure by 10 atm produce the same effect as cooling by 4 degrees, but fig.1d show cooling by 5-10 degrees. The seeming coincidence of maximum HSZ extension and maximum sea level during interglacials shown in fig.1f can be explained by delay of cooling wave with increase in depth. So, categorical statement that increase of pressure rather than cooling is a primary source of increase of HSZ volume for deep HB should be removed from abstract, conclusion and the end of chapter 2. It would be useful if authors present figure similar to their Fig.1f (and may be 1d, i.e. for HB=50m) but for experiment with prescribed change of pressure only with surface temperature fixed at -1.8C during 400 kyrs.*

We are very grateful for the reviewer for this comment. We agree that pressure changes can not cause onsets and disappearances of HSZ in our simulations. Only temperature changes may cause these onsets and disappearances. However, when the pressure and temperature at a given location in the the subsea sediments is close to a point at the HSZ stability curve, pressure changes may be important as well.

In particular, we made simulations similar to those presented in Sect. 2 of the manuscript, but imposing only temperature changes at the sediment-ocean water interface (accounting for oceanic transgressions and regressions) and neglecting changes of pressure due to sea level variations accompanying these transgressions and regressions. These simulations are alternative to the simulations suggested by the referee. We choose to perform the alternative simulations rather than those suggested by the referee because of the dominating impact of the temperature changes on the dynamics of subsea permafrost and HSZ: in the simulations suggested by the referee, subsea

permafrost and HSZ would not form at all.

In these simulations, for intermediate and deep HB, the timings of the HSZ onset are very close to their counterparts on the original simulations. The same is true for the HSZ bottom changes during the initial stage of its shoaling. Nevertheless, in the original simulations, HSZ survives much longer than in the simulations with neglected impact of pressure variations on HSZ. In particular, for $H_\mathrm{B} = 100$ m, HSZ exists until ≈74 kyr B.P. in the original simulations and only until ≈106 kyr B.P. in the additional ones.

In response to this comment, we extended our manuscript by the respective discussion in the end of Sect. 2 and by the supplementary Fig. S2 in which the results of the just mentioned simulations are shown. The statement in the text, which are listed by the referee, are clarified.

**Minor comments**

- *P.2, line 25-32. Why TB =-1.8C is not the same as Tf=-1C? This point should be explained.*
  It was an erroneous statement right before the phrase indicated by the referee that in our model the salinity of water in sediment pores is set equal to the oceanic one. Upon revision, this statement is corrected. Now we state that our model lacks the module for calculating the salinity of pore water, and the chosen $T_\mathrm{f} = -1^o$C value is one of typical values used in other simulations (Nicolsky et al., 2012; Portnov et al., 2014). In turn, the value of $T_\mathrm{B} = -1.8^o$C was adopted from (Romanovskii et al., 2005; Razumov et al., 2014). The corresponding references are added to the body of the text. We note that there is a scatter of these values among different papers. For instanse, Portnov et al. (2014) use $T_\mathrm{B} = -0.5^o$C, but only for a small area near the Yamal Peninsula. In turn, the value of $T_\mathrm{f} = -2^o$C

was used by Romanovskii et al. (2005). However, the precise values of $T_B$ and $T_f$ are of minor importance for the qualitative conclusions of our paper.

In addition, owing to $T_B$ is always smaller than $T_f$ in our simulations, a thin frozen layer survives in the upper part of the sediments even during interglacials. We acknowledge, that this may not be realistic. However, it hardly affects the main body of our results because we do not model the release of methane from hydrates. The respective discussion is added to Sect. 4.

- *P.2, line 32. Coefficient for specified initial linear temperature profile in K/m should be presented in the paper*
  It was an errouneous statement that the initial temperature profile in the sediments is linear. In the model, it is calculated as

$$T\left(z_j, t=0\right) = T\left(z_{j-1}, t=0\right) + G\left(z_j - z_{j-1}\right)/\kappa_j,$$

where $\kappa_j$ is a heat diffusivity which is equal either to $\kappa_f = 2.2$ W m$^{-1}$ K$^{-1}$ if a given computational level in the sediments is frozen or to $\kappa_u = 2.0$ W m$^{-1}$ K$^{-1}$ if it is unfrozen. The subscript $j$ indicates the computational level within the sediment (numbered from top to bottom), and $T\left(z_0, t=0\right) = T_B(t=0)$. The resulting $T(z, t=0)$ profile is close–to–linear with respect to the vertical coordinate with the coefficient which is either $2.7 \times 10^{-4}$ K m$^{-1}$ or $3.0 \times 10^{-4}$ K m$^{-1}$ depending on the state of the sediments.